# A Comprehensive Review of Receptor-Type Tyrosine-Protein Phosphatase Gamma (PTPRG) Role in Health and Non-Neoplastic Disease

**DOI:** 10.3390/biom12010084

**Published:** 2022-01-06

**Authors:** Christian Boni, Carlo Laudanna, Claudio Sorio

**Affiliations:** Department of Medicine, General Pathology Division, University of Verona, 37134 Verona, Italy; christian.boni@univr.it (C.B.); carlo.laudanna@univr.it (C.L.)

**Keywords:** PTPRG, phosphatase, inflammation, HCO_3_^−^ sensor, adhesion, interaction network, inflammation, central nervous system, neuropsychiatric disorders, Alzheimer disease, schizophrenia

## Abstract

Protein tyrosine phosphatase receptor gamma (PTPRG) is known to interact with and regulate several tyrosine kinases, exerting a tumor suppressor role in several type of cancers. Its wide expression in human tissues compared to the other component of group 5 of receptor phosphatases, PTPRZ expressed as a chondroitin sulfate proteoglycan in the central nervous system, has raised interest in its role as a possible regulatory switch of cell signaling processes. Indeed, a carbonic anhydrase-like domain (CAH) and a fibronectin type III domain are present in the N-terminal portion and were found to be associated with its role as [HCO_3_^−^] sensor in vascular and renal tissues and a possible interaction domain for cell adhesion, respectively. Studies on PTPRG ligands revealed the contactins family (CNTN) as possible interactors. Furthermore, the correlation of PTPRG phosphatase with inflammatory processes in different normal tissues, including cancer, and the increasing amount of its soluble form (sPTPRG) in plasma, suggest a possible role as inflammatory marker. PTPRG has important roles in human diseases; for example, neuropsychiatric and behavioral disorders and various types of cancer such as colon, ovary, lung, breast, central nervous system, and inflammatory disorders. In this review, we sum up our knowledge regarding the latest discoveries in order to appreciate PTPRG function in the various tissues and diseases, along with an interactome map of its relationship with a group of validated molecular interactors.

## 1. Introduction

The reversible phosphorylation in the different amino acid residues on proteins is deemed an essential mechanism for the cellular signal transduction. Living organisms through the balance between protein tyrosine kinase (PTK) and protein tyrosine phosphatase (PTP) activity orchestrate the levels of phosphotyrosine on proteins, allowing them to maintain cellular homeostasis. Alteration of the phosphoproteins profiles is associated with pathological conditions, and its description is instrumental for a mechanistic approach to healthy and diseased cells [1]. Historically, PTPs have generally been divided according to substrate specificity for tyrosine, serine, and threonine and grouped as protein tyrosine phosphatases (PTPs) and protein serine-threonine phosphatases (PSTPs) [2,3]. Protein tyrosine phosphatase gamma (PTPRG) is included in the R5 group, together with PTPRZ, and belongs to the subgroup of the family of receptor-like protein tyrosine phosphatases (RPTPs) [4,5]. This group of receptor proteins are composed of an extracellular region (ECD) that presents different structures. The extracellular domain consists of carbonic anhydrase-like domain (CAH), followed by a fibronectin-III domain, ideally implicated in cell–matrix and cell–cell interactions [6,7,8]. The phosphatase activity is assigned to the intracellular region (ICD), in the first membrane-proximal domain D1, followed by catalytically inactive D2 domain with regulatory and stability functions. The distinctive ICD-organization of RPTPs allows for a proposed regulation of the catalytic activity through a process of dimerization [9,10]. Similarly to other phosphatases, the structure of PTPRG also reveals a possible inactive dimeric form wherein electrostatic interactions, hydrogen bonds, and salt bridges are generated throughout dimer surface; in particular, the D2 domain interacts with the D1 of the other PTPRG molecule. The anti-parallel interaction between the domains causes an occlusion of the catalytic site, thus preventing the access of the substrate [11].

In this review, we focus on PTPRG and on collection of literature data present on the subject, dividing them according to the mechanism in which it is involved. We then summarize the known cellular mechanisms in which PTPRG is involved. Finally, we utilize eight specific targets on the basis of validated PTPRG interactors, derived from published data, to build a functional network, focusing on seven indexes of centrality representing a physiological regulatory sub-network involving PTPRG.

## 2. Signaling Pathways and Cellular Physiology Regulated by PTPRG

### 2.1. PTPRG Expression and Biochemical Features

PTPRG is ubiquitously expressed. It was originally identified in the central nervous system (CNS) [6] in parallel with the homologous PTPRZ [12,13]. mRNA and different protein isoforms of PTPRG were expressed in neuronal and accessory cells, including sensory pyramidal cells, microglia, and Purkinje cells [14,15,16]. PTPRG is also highly expressed in epithelial cells in various organs, including the genitourinary system (renal tubules), the gastrointestinal tract (large intestine), breast, and in the liver, particularly hepatocytes, with neuroendocrine cells in the gastrointestinal tract expressing the highest protein level [16]. Hematopoietic cells and precursors also express it [8,17,18,19]. Limited information is available regarding its biochemical feature. When overexpressed using baculovirus technology in Sf9 cells, it is expressed as a serine-phosphorylated, glycosylated polypeptide, and when expressed within the membrane bilayer, it appears to be allosterically activated by triphosphorylated nucleosides, including the non-hydrolyzable analog ATPγS. The physiological meaning remains unanswered [20], even if recent studies highlight regulatory roles to tryphosphorylated nucleotides: ATP and metformin reciprocally change cellular pH homeostasis in the liver, causing opposite shifts in liver activity of PTP1B, a key negative regulator of insulin signaling [21]. PTPRG was found to have the highest enzymatic activity in the peptide substrate panel and to form a stable dimer in solution, leading to the development of the a “head-to-toe” dimerization model that is distinct from the “inhibitory wedge” model and that provides a molecular basis for inhibitory regulation [11]. Furthermore, other evidence suggests that PTPRG might auto-dephosphorylate to the Y1307 residue in the D2 domain, indicating a potential regulatory site [22].

### 2.2. PTPRG Affects Cell Differentiation

Functional studies on PTPRG started with PC12D cells, where PTPRG was reported to block the growth of neurite during NGF (neuronal growth factor) exposure [23]. Genetic deletion of *Ptprg* in mice did not appear to grossly affect neuronal differentiation and adult neurogenesis [14], while a study performed in chicken spinal cord revealed a potential role of PTPRG in survival and localization/movement of motor neuron precursor (MNs) [14]. In this study, performed in chicken, PTPRG perturbation significantly reduced progenitor rates and neuronal precursor numbers, resulting in hypoplasia of neuroepithelium. PTPRG was suggested to interact with the WNT/β-catenin signaling and the modulation of β-catenin phosphorylation levels (based on in vitro dephosphorylation data) with the consequent influence of the TCF/LEF complex. Loss of function was associated to reduced adhesion and migration of precursor MNs [24]. Interestingly functional interaction with β-catenin was recently demonstrated in human chronic myeloid leukemia (CML) cells [25]. Since PTPs have been recognized as essential regulators of neuronal morphogenesis, research has focused both on intracellular phosphatase capacity and on potential interactors for the extracellular domain [26]. In murine embryonic stem cells, Ptprg was expressed in a limited timeframe from 14 to 18 days, suggesting an accurate regulation of PTPs during cell differentiation. Of note, its downregulation in embryonic stem (ES) cells is associated to complete lack of hematopoietic colonies [8]. Altogether, these data suggest an important regulatory role in key processes affecting embryonal development in different cellular models and species. Of note, the observation that mice lacking *Ptprg* apparently do not suffer of gross defects in neuronal and hemopoietic development in contrast to in vitro data showing that *PTPRG* perturbation affects cell differentiation might be explained by the development of alternative, compensatory, pathways during embryonic development, likely involving other members of the wide PTP family. No data supporting this hypothesis are currently available, and thus further experimental studies are warranted to fill this gap.

### 2.3. Cell Adhesion

The growth factor pleiotrophin (PTN) and contactin (CNTN1) are known to be able to bind the extracellular domain (ECD) of the homologous PTPRZ [27]. The six members of contactin (belonging to Ig-CAMs subfamily) consist of Ig-like repeats and fibronectin III-like domains coupled with a GPI (glycophodphatidylinositol) molecule that anchor it to external leaflet of the cell membrane. These proteins are involved in organization of axonal domains and guidance, neuronal- and neuritogenesis, synapse formation, and plasticity whose interaction with PTPRZ may regulate signals during neuronal development [28,29]. Additionally, some members of this family (CNTN3–6) may be also potential ligands for the CAH domain of PTPRG [30]. Indeed, structural and biochemical investigations have shown an interaction with CNTN3–6, particularly CNTN3, which forms complexes with *ptprg* both in cis, on the same membrane and in trans when they are expressed on membranes of different rod photoreceptors of the murine retina [31]. Although these molecules have 40–60% sequence similarity, Mercati et al. identified the same strong association with PTPRG among CNTN4-5–6 [32]. Furthermore, silencing either CNTN6 or *ptprg* deficiency, both expressed in the V layer of the cerebral cortex, alters motor coordination in mouse models during specific (road walking and string) tests [14,33]. These findings suggest the importance of interaction among these partners and a role of PTPRG in acquiring proper motor function. In addition, receptor protein tyrosine phosphatases (RPTPs) are regarded as neuronal receptors that take part in the cell adhesion process on the basis of the presence of cell adhesion-related domain on the ECD and for the ability to regulate the action of tyrosine kinases [26]. Of note, dimerization is associated to inhibition of RPTP activity [11,34,35]. However, there is no evidence that this phenomenon occurs when both PTPRG and PTPRZ interact with CNTN partners, suggesting that these interactions might guide these PTPs to the proper action site [30,36].

A link with adhesion processes was also found in leukocytes. A large family of heterodimeric cell surface receptors called integrin drives cell-to-cell and cell-to-matrix adhesion and migration mechanisms [37]. The use of a new approach based on cell penetrating peptides (CPPs) coupled to a computationally identified sequence corresponding to the wedge domain (WD) led to the transition to PTPRG monomer, unleashing the phosphatase activity and preventing, in human primary monocytes, integrin activation triggered by chemoattractants mediating rapid arrest in underflow adhesion assays. Importantly, P1-WD treatment inhibits the transition to high-affinity state of LFA-1, promoted by fMLP stimulation, through JAK2 dephosphorylation [38]. The same approach was utilized in healthy B-lymphocytes compared to chronic lymphocytic leukemia cells (CLL) where PTPRG activation was found to inhibit adhesion by preventing the LFA-1 high-affinity state transition induced by CXCL12 or BCR necessary to bind on both ICAM-1 and VCAM1. Activated PTPRG decreases the activity of JAK2 and BTK kinases involved in chemokine signaling to β1 and β2 integrin activation. Furthermore, the downstream effector of both kinases, the GTPase RhoA, is also compromised by P1-WD pretreatment, confirming the key role of PTPRG in the leukocyte adhesion dynamic [39,40]. Importantly, and specifically in the CLL context, activation of PTPRG triggered rapid and robust apoptosis, whereas normal B cells were unaffected. This raises the possibility that PTPRG could be a tumor suppressor gene in CLL.

### 2.4. PTPRG as HCO_3_^−^ Sensor

Carbonic anhydrases are a family of enzymes involved in the interconversion of HCO_3_^-^ and CO_2_ [41]. The CAH-like domain of PTPRG-ECD lacks key histidine residues essential for catalytic activity [6], suggesting a non-catalytic role of this domain. Involvement in the regulation of acid–base balance was described in renal proximal tubules (PTs) in *ptprg*^–*/*–^ mice, emphasizing the loss of ability to counteract the basolateral [HCO_3_^−^] changes, compared to WT. In addition, in a simulated metabolic acidosis *Ptprg* interferes with the restoration of the arteriolar pH compared to WT, showing a substantial acidosis in *ptprg*^–*/*–^ mice [42]. In supporting the involvement of *Ptprg* as CO_2_/HCO_3_^−^ sensor, the same role was also confirmed at the arteriolar level in *ptprg*^–*/*–^ mice. Acid–base perturbations in an equilibrium model produce variations in at least two of these main parameters: pH_o_, [HCO_3_^−^]_o_, and pCO_2_. Through out-of-equilibrium (OOE) studies, researchers have demonstrated that endothelial *Ptprg* affects the vasomotor effects induced by CO_2_/HCO_3_^−^ by regulating the Ca^2+^-sensitivity in arteriolar VSMC (vascular smooth muscle cells) that might enhances intracellular [Ca^2+^] in endothelial cells by implementing NO synthesis and endothelium-dependent hyperpolarization (EDH)-type responses. Thus, *Ptprg* was suggested to exert HCO_3_^−^ sensor function and to regulate the endothelium-dependent vasorelaxant effects [43,44].

## 3. Participation of PTPRG in Different Pathological Processes

The first recurrent evidence regarding the role of PTPRG were related to genetic and epigenetic alterations, including non-random deletions, loss of heterozygosity, and alteration of the CpG islands methylation profile identified in several types of cancer including sarcoma, carcinomas, and leukemia/lymphomas [45,46,47]. On the basis of this evidence and other evidence, PTPRG has been recognized as a tumor suppressor gene, and numerous findings report an impaired tumorigenic after introduction or re-expression of PTPRG [48,49,50]. A comprehensive review covering the multiple roles of PTPRG in cancer is currently in press [51].

### 3.1. Neuropsychiatric and Behavioral Disorders

Studies on the role of PTPs in the CNS have identified them as mediators of differentiation and synaptic organizers. Furthermore, they can take part during the modulation of the processes related to learning and memory [52,53]. Therefore, progress in identifying genes associated to neuropsychiatric pathologies could provide insights into etiology and the search for molecular targets [54,55]. A clear example was the work of Harroch’s group, which using *ptprg^–/–^* and its close paralogue *ptprz^–/–^* mice identified specific behavioral changes in the animals. Intriguingly, the loss of combined or individual phosphatase was associated with a specific neurotransmitter modulation in distinct areas of the brain, indicating these as appealing therapeutic targets [56]. In the present case, PTPRG has a precise distribution in the brain and is characteristic of a group of neurons [14,15]. Its high affinity for contactin members, which are involved in autism spectrum disorder (ASD), suggest a potential role in this disease [57,58]. Various evidence indeed suggesting PTPRG as a susceptible gene involved also in neuropsychiatric disorders. Whole-genome sequencing analysis in several cohorts of schizophrenic patients revealed mutations involving five genes, including PTPRG. Missense and nonsense mutations in PTPRG (I837S and R422X, leading to a loss of phosphatase function) have been reported, suggesting phosphorylation imbalance as a possible cause of developmental neurodisorders [59,60]. Furthermore, the presence of some polymorphisms and intron variants would be associated for bipolar schizoaffective disorder [61,62,63]. Likewise, PTPRG has been associated with Alzheimer’s disease (AD), one of the most frequent forms of dementia in the world [64]. The predominant form of Alzheimer’s diagnosed is late onset (LOAD), in which the strongest risk factor considered remains APOE-ε4. A single-nucleotide polymorphism (SNP) identified on PTPRG has been identified after a family-based genome-wide association (GWAS) and meta-analysis. The rs7609954 variant achieves high genome-wide significance pointing to PTPRG and its phosphatase regulatory functions as a possible risk gene for LOAD [65].

### 3.2. Inflammation

The countermeasures adopted by the organisms such as immunity and inflammation are adaptive mechanisms triggered by toxic stimuli (infection, tissue injury, etc.), which work to recover the homeostatic state [66]. Deficiencies or excessive immune responses trigger various pathological conditions. Similarly, perpetrated inflammatory response also produces a chronic response that is often the cause of several pathologies [67]. The study of PTPRG in the CNS (central nervous system) of rat brain unearthed different PTPRG isoforms derived from alternative splicing. Although all isoforms were well represented, higher expression of the soluble protein form (sPTPRG), constituted by the 120 KDa ECD, was found during the embryonic stages [68]. Other RPTPs can be processed by targeted proteolytic mechanisms by detaching the ECD from the intracellular phosphatase domain [69,70,71]. In mice treated with LPS for 24 h, a general inflammatory state was induced and was associated with increased expression of all PTPRG isoforms in cortical astrocytes, particularly the 120 KDa soluble isoform. In 5XFAD Alzheimer’s disease mouse model, astrocytes surrounding the β-amyloid plaques are highly positive for PTPRG expression, suggesting as a possible factor for astrocytic activation during neuroinflammation [15]. Furthermore, this processed isoform (sPTPRG) has been shown to be present in both human and murine liver and serum. The hepatocyte-derived HepG2 cell line shows a reduction of sPTPRG expression when treated with metalloprotease inhibitors (Furin inhibitor and Ilomastat); meanwhile, a toxic molecule such as ethanol (50 mM) produces the rise of sPTPRG form in the media, suggesting a regulatory mechanism of this isoform, possibly mediated by an enzymatic cleavage. These data led to the analysis of primary serum samples from healthy donors and patients with elevated alanine aminotransferase (ALT), an established biomarker of liver injury [15,72]. In serum samples from liver inflammation, Moratti et al. demonstrated the enhancement of soluble PTPRG compared to healthy donor samples, proposing PTPRG as a putative biomarker of liver injury [73]. Similarly, metalloproteases were able to release the membrane-associated portion of PTPRZ, while the presenilin/γ-secretase induced the release of the ICD fragment that was detected both in the cytosol than in the nucleus [74]. In addition, analysis of cerebrospinal fluid (CSF) identified sPTPRZ as a biomarker of brain tumors [75]. Involvement in metabolic disorders has also been supported by published data. Chronic inflammatory processes are often found in overweight or obese individuals. Several complications have arisen from this condition: insulin resistance, diabetes type 2 (T2DM), liver and cardiovascular disease, and cancer [76,77]. Given the established role of PTPs as kinase modulators, a role of phosphatases as insulin receptor (IR) inactivators was proposed [78]. Bottini’s group identified small-molecule inhibitor of LMWPTP (low-molecular-weight tyrosine phosphatase) capable of blocking insulin resistance and T2DM occurrence [79]. Instead, in humans, liver PTPRG overexpression appears to correlate with inflammatory mechanisms and obesity [80]. The study of a Korean population using an integrated predictive model of T2DM that combined a purely clinical component with a genetic component based on SNP-point score highlighted five main SNPs. Among these, rs9311835 was identified on the PTPRG intron, increasing its interest as a predictive sign of T2DM [81]. Molecular mechanisms analysis performed on Ptprg^–/–^ mice showed a robust improvement in glucose metabolism induced by the increased insulin sensitivity in hepatocytes, thus remaining protected from the development of hyperglycemia when compared to Ptprg^+/+^. Both insulin signaling and glucose metabolism were affected by PTPRG through modulation of the phosphorylation of some substrates, including IR, P85A, and AKT. Indeed, hepatic insulin resistance was found in Ptprg-overexpressing mice at values comparable to obese subjects. Therefore, inflammatory states such as obesity are able to generate an NF-kB-induced Ptprg expression mechanism due to a kB-binding site identified on the Ptprg promoter [80]. Although many aspects are to be clarified, an early study demonstrates a consistent increase in PTPRG in the GDM (gestational diabetes mellitus) group compared with the control, identifying it as a possible disease biomarker [82]. Excessive inflammatory response, oxidative stress-induced damage, and aging are some of the major causes involved in the progression of NAFLD (nonalcoholic fatty liver disease) to fibrosis and cancers [83]. NLRP3 inflammosome (NOD-type receptors 3) is activated under these conditions by inducing apoptosis and the production of pro-inflammatory cytokines. Studying the KO of the NLRP3 inflammation in liver tissues of young and old mice, Gallego et al. showed a reduction in the activity of SOD (super-oxide dismutase), correlating it with the age of the mouse and a reduction in the expression of Ptprg, confirming its role as a biomarker of liver damage [84].

### 3.3. Other Diseases

Despite several PTPRG-regulated processes being well established and constantly explored, its functional correlation in other pathologies is either tenuous or totally unknown. Acute kidney injury (AKI) is a kidney disorder characterized by a heterogeneous group of dysfunctions such as decreased glomerular filtration rate with and oliguria with a variable etiology [85]. Recently, the SARS-CoV-2 pandemic has considerably increased the occurrence of AKI, especially in patients with comorbidities [86]. The study of biomarkers such as miRNAs in the various stages of AKI suggested a possible prediction profile of recovering/non-recovering patients after 90 days. A screening on different miRNAs in AKI patients has reported the upregulation of miR-141, which both in a model of H_2_O_2_-induced tissue damage or in samples disease is able to negatively modulate PTPRG mRNA expression. The downregulation of PTPRG could therefore be fundamental for the acceleration of renal fibrotic processes through EGFR hyper-stimulation, formerly known as PTPRG substrate [87]. As PTPRG is known to have a subtle regulation by small non-coding RNAs [88,89], Liu et al. analyzed the cardioprotective capacity of miR-208a deemed a heart attack biomarker. In particular, miR-208a knockdown was able to reduce ROS intracellular levels in cardiomyocytes by increasing the SOD (superoxide dismutase) and CAT (catalase) enzymes in the culture medium. The miRNA also targeted two PTPs by reducing the expression of PTPRG and PTPN4, altering the apoptotic ratio and acting as a moderator of the expression of ROS-reducing enzymes [90]. Degenerative disease named Fuchs’ endothelial dystrophy (FED) can affect the corneal endothelium, generating microscopic outgrowths (guttae) in the corresponding basal lamina. A more common late-onset form and a rarer early onset form can be distinguished [91]. Although a strong correlation with the disease was found for some genes (TCF4, TCF8, AGBL1, LOXHD1, SLC4A11, and others), the study of PTPRG on several SNPs (rs7640737 and rs10490775) appear conflicting, showing an association without reaching a genome-wide significance in FED [92,93,94,95]. However, the data reported for each disease constitute a starting point and require confirmation/further investigation to confirm the involvement of PTPRG. The pathological roles of PTPRG phosphatase were collected in Table 1.

## 4. PTPRG Network’s Analysis

Current data describe the role of PTPRG in various non-neoplastic contexts analyzed in this review (Figure 1). Signal transduction pathways depend on highly coordinated flux of information within the cell mediated in a large part by protein–protein interactions. As such, the identification of reliable interatomic maps is instrumental to the in silico reconstruction of a reliable signaling network governed by a specific molecular target. We observed that different interatomic databases report tens or hundreds of PTPRG interactors, with only a few being cross-validated by means of several, different, methodologies. To focus on more reliable and consistent data, we focused on manually curated data, on the basis of the availability of several independent and published pieces of evidence previously described. This approach prompted focusing the network inference on eight PTPRG direct targets: JAK2, BTK, ABL1, AXL, CTNNB1, BCR, EGFR, and FGFR1. These targets have been considered as bioinformatic probes in order to extract the sub-network representing the PTPRG-specific interatomic space from a large human interactome dataset. Network inference generated a first neighbor (FN) network of 2976 nodes (interactors) connected by 123,961 binary, non-redundant, un-directed edges. This was a huge informational space, suggesting that PTPRG may affect many signaling pathways and biological processes at once, in line with wide tissue distribution of this PTP. To train such complexity, as well as to be able to categorize the most important nodes (interactors), we calculated seven centrality indexes, differently indicative of the topological and functional node relevance in the global network. Figure 2 shows the statistical distribution (as violin plots) of the values of the different indexes calculated for every node. To allow direct comparison, the log10 for every value was calculated. We can appreciate that the distribution for closeness, eccentricity and radiality is almost completely flat, indicating that all nodes in the network do show almost the same value around the median. Since closeness, eccentricity and radiality calculate the overall “dispersion” of information in a network, these data suggest that no nodes are really isolated or marginal. This may suggest a high compactness of the networks and the overall capability of the PTPRG network to generate integrated signaling modules processing information in quite a coordinated way. This observation was in keeping with the distribution of the other five centralities. Here, we can observe a quite symmetric distribution of the values, but with clear morphologies (patterns) emerging in the plot. Most of the nodes do show values evenly distributed around the median (red line) or the quartiles (blue lines). However, very few nodes aggregate to small, highly dense, clusters, evidenced in the plots by top circles. Of particular interest are the distributions of betweenness and centroid, which indicate the capability of connecting distant nodes (betweenness) and highly dense clusters (centroid). These two parameters clearly suggest that few nodes in the network are highly involved in information flow processing and signal coordination and integration. The emergence of a restricted group of highly interacting, regulatory nodes (signaling proteins) is also shown in the multidimensional plot shown in Figure 3, where the distribution according to four value/node is shown. Here, the top circle highlights a group of nodes with high values for betweenness, centroid, degree, and bridging centralities, all indexes indicating effective information processing. By applying mathematical filters to the network (by using an AND Boolean operator), we then extracted the most relevant nodes, all characterized by values of betweenness, centroid, degree, and bridging centralities over the average.

This procedure allowed for the extracting of a sub-network of 24 nodes and 72 interactions (Figure 4). This sub-network possibly represents a highly relevant regulatory signaling module in PTPRG physiology. Finally, ontology and statistical analysis were applied to identify the most represented signaling pathways and pathological processes regulated by the sub-network in Figure 4. Analysis was done in the context of KEGG, Reactome, and GO databases and shows the node participation to individual pathways (Table 2).

## 5. Conclusions

The study of PTPs represents a challenging field as less experimental tools are available for most of these targets in comparison to cognate tyrosine kinases, as an example. However, considerable progress did stratify over the years, leading to the identification of relevant targets whose biological meaning is being investigated in deeper detail. Adding molecular details is instrumental to the development of possible clinical applications. Indeed, the recognition of a tumor suppressor (TS) role in many different cellular models underline a potential application of a TS re-expression/activation that confirmed a strong therapeutic potential of this strategy. Here, we have described the multiple roles played by PTPRG in various physiopathological conditions. Considering PTPRG formed by different intra- and extracellular domains [6,68], it participates in complex physiological processes such as cell differentiation and leukocyte–endothelial adhesion, and in specific tissues it is able to detect the concentration of CO_2_/ HCO_3_^−^ [8,39,43]. Although the distinctive feature of PTPRG is that of TS in many forms of cancer, it also participates in other pathological processes such as liver inflammation, diabetes, and neuropsychiatric/behavioral disorders [56,73,80]. Other roles are only sketched, but some recent evidence has linked PTPRG with renal (e.g., AKI) and corneal (e.g., FED) dysfunctions.

PTPRG multiple roles are supported by the result of our bioinformatics analysis, performed on carefully cured data available from the literature, which summarize current findings and suggest future avenues of research. Indeed, in this unbiased analysis, known pathways emerged (chronic myeloid leukemia, acute leukemia, B cell receptor pathway, long-term depression, and leukocyte trans endothelial migration in KEGG as an example) but for others such as oocyte maturation, FCepsilonR, renal reabsorption, and Chagas disease, there is no currently published evidence and thus these areas may deserve future investigations.

## Figures and Tables

**Figure 1 biomolecules-12-00084-f001:**
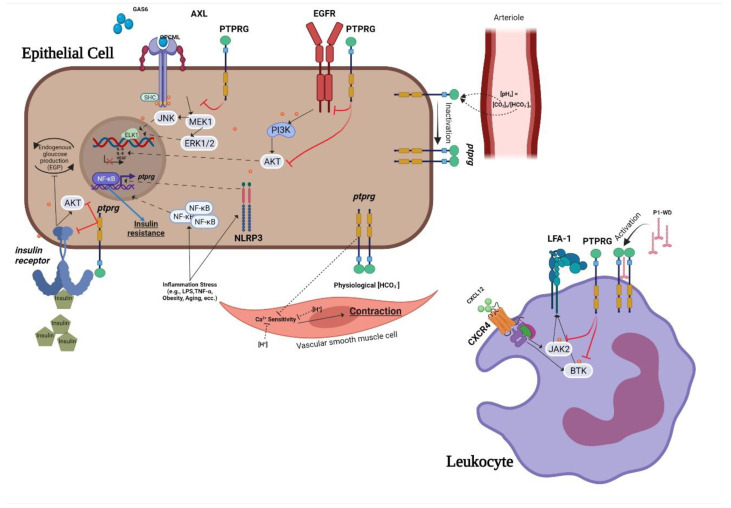
Summary of the role of PTPRG in physiopathology of non-neoplastic cell.

**Figure 2 biomolecules-12-00084-f002:**
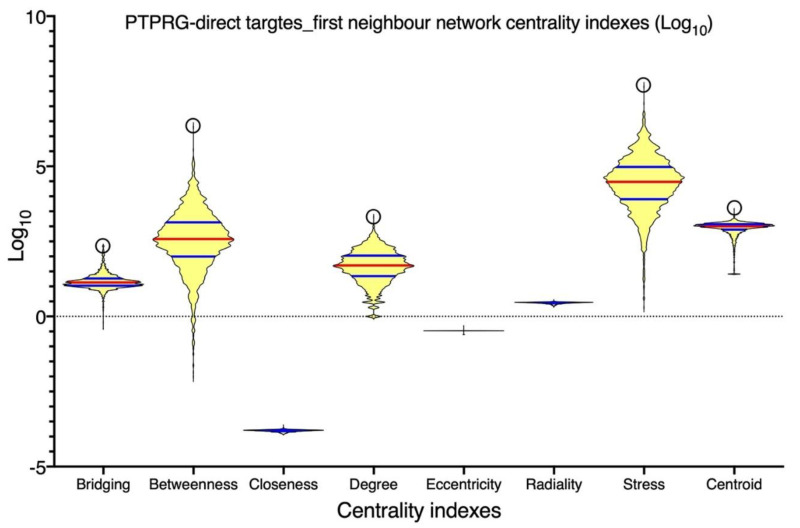
Node centrality indexes computational analysis of the first neighbor’s protein–protein interaction (PPI) network inferred upon the eight known PTPRG direct targets: JAK2, BTK, ABL1, AXL, CTNNB1, BCR, EGFR, and FGFR1. Shown are violin plots of the statistical distribution of the values of the different node centrality indexes calculated for every node (2976 nodes, connected by 123,961 non-redundant, un-directed edges). To allow direct comparison, the log10 for every value was calculated. The upper open circle indicates the position of PTPRG. Red lines are medians, blue lines are quartiles.

**Figure 3 biomolecules-12-00084-f003:**
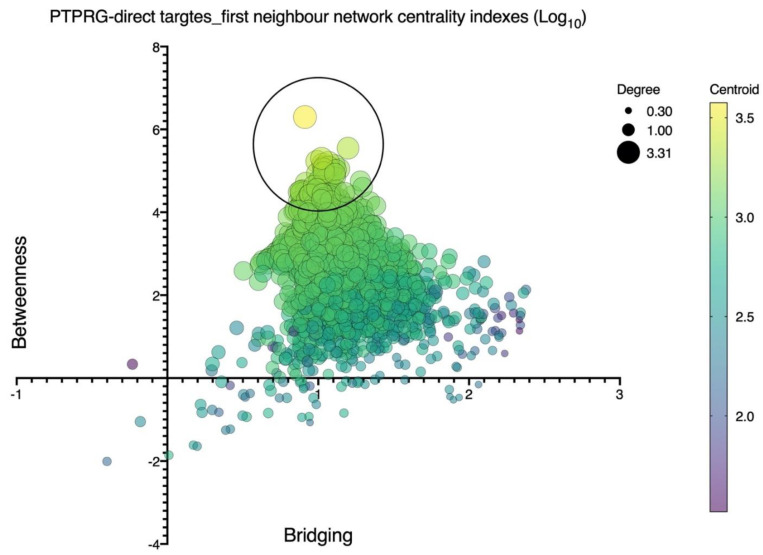
Node centrality index computational analysis of the first neighbor’s protein–protein interaction (PPI) network inferred upon the eight known PTPRG direct targets: JAK2, BTK, ABL1, AXL, CTNNB1, BCR, EGFR, and FGFR1. Shown is multidimensional data plotting, including *x*–*y* position (betweenness and bridging), size (degree), and color (centroid). Distribution is according to these four value/nodes. The large top open circle highlights a group of nodes with higher concurrent values of betweenness, centroid, degree, and bridging centralities, indicating effective information processing.

**Figure 4 biomolecules-12-00084-f004:**
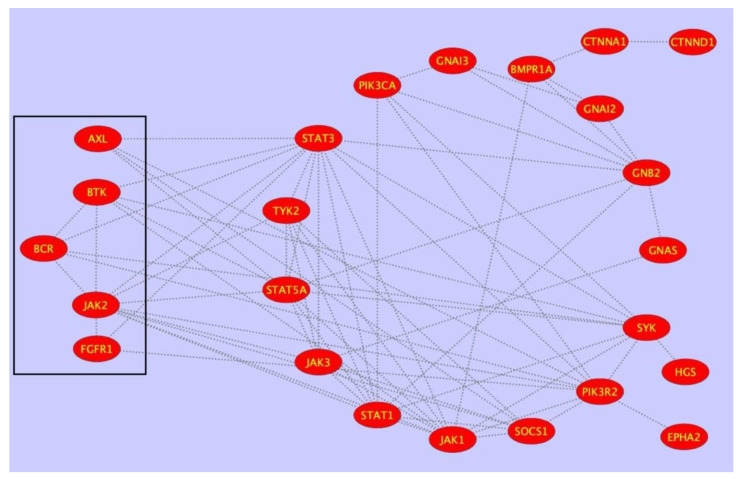
Sub-network of 24 nodes and 72 interactions extracted by applying mathematical filters to the PTPRG-specific first neighbor (FN) PPI network of 2976 nodes (interactors) connected by 123,961 binary, non-redundant, un-directed edges. The sub-network was generated by using OR and AND Boolean operators and allowed extracting the most topologically relevant nodes, all characterized by values of betweenness, centroid, degree, and bridging centralities over the average. This sub-network possibly represents a highly relevant regulatory signaling module in PTPRG physiology.

**Table 1 biomolecules-12-00084-t001:** Summary table of the pathological processes where PTPRG is involved.

Disease	Role of PTPRG	References
Cancers	Tumor suppressor role in many malignancies with a reported oncogenic role in glioblastoma	[51]
Neuropsychiatric and behavioral disorders	PTPRG variants have been associated to neuropsychiatric pathologies such as autism and schizophrenia	[56,59,60,96,97]
Inflammation	At the hepatic level, the overexpression of PTPRG correlates with inflammatory processes especially in obese subjects. Increased levels of the ECD portion in serum constitutes a potential biomarker of liver damage.	[73,80,81,82,84]
Acute kidney injury	PTPRG shutdown by mir-141 is associated to accelerated fibrotic process in AKI-prone kidney.	[87]
Fuchs’ endothelial dystrophy	Possible correlation between FED and *PTPRG*	[93]

**Table 2 biomolecules-12-00084-t002:** Ontology and statistical analyses were applied to identify the most represented signaling pathways and pathological processes regulated by the sub-network described in Figure 1. Analyses were done in the context of KEGG, Reactome, and GO databases and show the node participation to individual pathways. Functional relevance is proportional to the XD-score and q-values.

Pathway or Process (KEGG)	XD-Score	q-Value	Overlap/Size
Acute myeloid leukemia	0.24613	0.00665	4/52
Jak-STAT signaling pathway	0.23695	0.00000	10/134
Pancreatic cancer	0.22415	0.00177	5/70
Type II diabetes mellitus	0.21751	0.03097	3/43
Fc epsilon RI signaling pathway	0.18459	0.01242	4/65
Leukocyte transendothelial migration	0.18334	0.00111	6/98
Primary immunodeficiency	0.18086	0.11451	2/33
Endometrial cancer	0.17844	0.04415	3/50
B cell receptor signaling pathway	0.17032	0.01248	4/69
Chronic myeloid leukemia	0.17032	0.01248	4/69
Chemokine signaling pathway	0.15020	0.00003	9/170
Long term depression	0.14897	0.05952	3/57
Aldosterone-regulated sodium reabsorption	0.14897	0.13898	2/38
Progesterone-mediated oocyte maturation	0.14097	0.01893	4/79
Chagas disease	0.14046	0.00665	5/99
**Pathway or Process (REACTOME)**	**XD-Score**	**q-Value**	**Overlap/Size**
G ALPHA Z SIGNALLING EVENTS	0.63202	0.05139	2/12
ADP SIGNALLING THROUGH P2Y PURINOCEPTOR 12	0.59693	0.01136	3/19
DOWN STREAM SIGNAL TRANSDUCTION	0.55359	0.00022	5/34
G PROTEIN ACTIVATION	0.44535	0.02116	3/25
TIE2 SIGNALING	0.43594	0.09034	2/17
SIGNAL AMPLIFACTION	0.37915	0.02764	3/29
COLLAGEN MEDIATED ACTIVATION CASCADE	0.34631	0.12357	2/21
PI3K CASCADE	0.34035	0.03184	3/32
GAB1 SIGNALOSOME	0.32899	0.64986	1/11
GS APLHA MEDIATED EVENTS IN GLUCAGONE SIGNALLING	0.32899	0.12357	2/22
PLATELED ACTIVATION TRIGGERS	0.32899	0.00131	5/55
**Pathway or Process (Go Data base)**	**XD-Score**	**q-Value**	**Overlap/Size**
1-phosphatidylinositol-3-kinase activity	0.77762	0.01530	2/10
Insulin receptor substrate binding	0.59301	0.02376	2/13
Non-membrane spanning protein tyrosine kinase activity	0.57762	0.00000	6/40
G-protein beta/gamma-subunit complex binding	0.57762	0.00119	3/20
Phosphatidylserine binding	0.37762	0.54997	1/10
Cadherin binding	0.37762	0.04768	2/20
Peptide hormone receptor binding	0.37762	0.54997	1/10
Guanyl nucleotide binding	0.35262	0.00443	3/32
RNA polymerase II core promoter sequence-specific DNA binding	0.34126	0.05334	2/22
Insulin-like growth factor receptor binding	0.31096	0.55963	1/12
Gamma-catenin binding	0.31096	0.55963	1/12

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
