# Peer review of "A Comprehensive Review of Receptor-Type Tyrosine-Protein Phosphatase Gamma (PTPRG) Role in Health and Non-Neoplastic Disease"

_biomolecules, 2022, doi:10.3390/biom12010084_

Round 1

Reviewer 1 Report

The authors are presenting the review of PTPRG roles in an organized manner. I would like to recommend accepting this review for the journal.

Minor corrections for the typos are needed such as HCO3- --> HCO3-, Raectome --> Reactome, and Reactome e Go --> Reactome and GO.

This review is presenting the physiological roles of PTPRG related to diseases, such as cancer, inflammation, autism, and so on. This topic is relevant in the field of protein phopshatases, so it is an interesting subject in the scientific society. I am not able to find any problems in the subject area, the references, the tables and figures.

The conclusions are well summarized with the main contents. If possible, I would like to recommend adding the biochemical aspects of PTPRG. This will be more comprehensive review.

Author Response

The authors are presenting the review of PTPRG roles in an organized manner. I would like to recommend accepting this review for the journal.

Minor corrections for the typos are needed such as HCO3- --> HCO3-, Raectome --> Reactome, and Reactome e Go --> Reactome and GO.

Done

This review is presenting the physiological roles of PTPRG related to diseases, such as cancer, inflammation, autism, and so on. This topic is relevant in the field of protein phopshatases, so it is an interesting subject in the scientific society. I am not able to find any problems in the subject area, the references, the tables and figures.

 Thank you for your endorsement

The conclusions are well summarized with the main contents. If possible, I would like to recommend adding the biochemical aspects of PTPRG. This will be more comprehensive review.

Thank you for the useful suggestion, we added a paragraph summarizing the biochemical data available.

Reviewer 2 Report

The  review  “ A comprehensive review of Protein Tyrosine Phosphatase Gamma (PTPRG) role in health and non-neoplastic disease” presented by Christian Boni, Carlo Laudanna  and Claudio Sorio  is well written, the introduction is clear, the aims and the hypothesis are correctly formulated.

The article contains comprehensive information on the properties of this enzyme, starting from the mechanism and localization of its expression and the effect on cellular processes such as cell differentiation, adhesion etc. Of particular interest is the section describing participation of PTPRG in different pathological processes. This section presents the mechanisms of PTPRG involvement in neuropsychiatric and behavioral disorders and Inflammation. Despite several PTPRG regulated processes are well established and constantly explored, its functional correlation in other pathologies is either tenuous or totally unknown. For example, acute kidney injury, degenerative disease named Fuchs’ endothelial dystrophy (FED). However, the data reported for each disease constitute a starting point and require confirmation/further investigation to confirm the involvement of PTPRG. The PTPRG network's analysis is of particular value in the review. Current data are described the role of PTPRG in various non-neoplastic contexts analyzed in this review. All results are graphically presented in Figure 1-4 and in Table 2. The study of PTPs represent a challenging field as less experimental tools are available for most of these targets in comparison to cognate tyrosine kinases, as an example.

The authors cite 91 literature sources since 1991. However, it pays attention. that the cited studies in 2020-2021 are few. Perhaps, the peer-reviewed review will be able to draw attention to the development of this interesting direction.

Minor comments:

  1. I propose to make a slight change to the title. If I could suggest "A comprehensive review of Receptor-type Tyrosine-Protein Phosphatase Gamma (PTPRG) role in health and non-neoplastic disease", but the final decision is obviously on charge of the authors.
  2. Lines 286-288 probably contain the comments of the co-author: “section may be divided by subheadings. It should provide a concise and precise description of the experimental results, their interpretation, as well as the experimental conclusions that can be drawn”. They need to be removed.

After a little editing, the article can be published.

Reviewer

Author Response

The article contains comprehensive information on the properties of this enzyme, starting from the mechanism and localization of its expression and the effect on cellular processes such as cell differentiation, adhesion etc. Of particular interest is the section describing participation of PTPRG in different pathological processes. This section presents the mechanisms of PTPRG involvement in neuropsychiatric and behavioral disorders and Inflammation. Despite several PTPRG regulated processes are well established and constantly explored, its functional correlation in other pathologies is either tenuous or totally unknown. For example, acute kidney injury, degenerative disease named Fuchs’ endothelial dystrophy (FED). However, the data reported for each disease constitute a starting point and require confirmation/further investigation to confirm the involvement of PTPRG. The PTPRG network's analysis is of particular value in the review. Current data are described the role of PTPRG in various non-neoplastic contexts analyzed in this review. All results are graphically presented in Figure 1-4 and in Table 2. The study of PTPs represent a challenging field as less experimental tools are available for most of these targets in comparison to cognate tyrosine kinases, as an example.

The authors cite 91 literature sources since 1991. However, it pays attention. that the cited studies in 2020-2021 are few. Perhaps, the peer-reviewed review will be able to draw attention to the development of this interesting direction.

Actually, most of the more recent works concentrate on cancer.

Minor comments:

  1. I propose to make a slight change to the title. If I could suggest "A comprehensive review of Receptor-type Tyrosine-Protein Phosphatase Gamma (PTPRG) role in health and non-neoplastic disease", but the final decision is obviously on charge of the authors.

Thank you, accepted

  1. Lines 286-288 probably contain the comments of the co-author: “section may be divided by subheadings. It should provide a concise and precise description of the experimental results, their interpretation, as well as the experimental conclusions that can be drawn”. They need to be removed.

Removed

After a little editing, the article can be published.

Reviewer

Thank you for your endorsement